# External gas-assisted mold temperature control and optimization molding parameters for improving weld line strength in polyamide plastics

**Nguyen Truong Giang**[1,2], **Pham Son Minh**[3], **Tran Anh Son**[1,2], **Van-Thuc Nguyen**[3], **Tran Minh The Uyen**[3], **Thanh Trung Do**[3], **Van Thanh Tien Nguyen**[4]*

**1** Ho Chi Minh City University of Technology (HCMUT), Ho Chi Minh City, Vietnam, **2** Vietnam National University Ho Chi Minh City, Ho Chi Minh City, Vietnam, **3** HCMC University of Technology and Education, Ho Chi Minh City, Vietnam, **4** Industrial University of Ho Chi Minh City, Ho Chi Minh City, Vietnam

* thanhtienck@ieee.org

**Data Availability Statement:** All relevant data are within the paper.

## Abstract

In this study, we present a novel approach to injection molding, focusing on the strength of weld lines in polyamide 6 (PA6) composite samples. By implementing a mold temperature significantly higher than the typical molding practice, which rarely exceeds 100°C, we assess the effects of advanced mold temperature management. The research introduces a newly engineered mold structure specifically designed for localized mold heating, distinguishing it as the 'novel cavity.' This innovative design is compared against traditional molding methods to highlight the improvements in weld line strength at elevated mold temperatures. To optimize the molding parameters, we apply an Artificial Neural Network (ANN) in conjunction with a Genetic Algorithm (GA). Our findings reveal that the optimal ultimate tensile strength (UTS) and elongation values are achieved with a filling time of 3.4 seconds, packing time of 0.8 seconds, melt temperature of 246°C, and a novel high mold temperature of 173°C. A specific sample demonstrated the best molding parameters at a filling time of 3.4 seconds, packing time of 0.4 seconds, melt temperature of 244°C, and mold temperature of 173°C, resulting in an elongation value of 582.6% and a UTS of 62.3 MPa. The most influential factor on the PA6 sample's UTS and elongation at the weld line was found to be the melt temperature, while the filling time had the least impact. SEM analysis of the fracture surfaces revealed ductile fractures with rough surfaces and grooves, indicative of the weld line areas' bonding quality. These insights pave the way for significant improvements in injection molding conditions, potentially revolutionizing the manufacturing process by enhancing the structural integrity of the weld lines in molded PA6 samples.

## 1. Introduction

In the micro and thin wall injection molding process, the melted plastic is rapidly cooled down due to the high ratio of surface-to-volume [1–4]. The viscosity of the injection plastic rises,

**Funding:** The author(s) received no specific funding for this work.

leading to a higher fluid's resistance to flow during the molding process. To solve this problem, improving the injection pressure could increase the filling rate. However, it will cause inhomogeneous pressure distribution. Therefore, the density and the shape of the injection sample also oscillate along the flow path. Besides, increasing the mold temperature by using a mold temperature control system is also a solution to improve the filling rate by slowing down the plastic cooling process. For example, Roth et al. [5] used a dynamic mold temperature control system to inject a thin wall plate. The plate aspect ratio is increased by 125% and the thickness deviation is only 3%. Chen et al. [6] also applied a mold temperature control system to generate a rubber product in the microcellular injection molding process. The surface roughness of the sample has a 65% improvement compared to without using the mold temperature control system. Dong et al. [7] proved that using a dynamic mold temperature control system help remove the surface bubble in the microcellular injection molding. The reason for this phenomenon is that the gas is dissolved in the melted plastic when it has a higher temperature. Kurt et al. [8] investigated the impacts of cavity pressure and mold temperature during the injection molding operation. Increasing the cavity pressure and mold temperature results in a lower rate of shrinkage in both x- and y-directions.

A weld line in injection molding is generated when two molten polymers stream meat in opposite directions, as shown in Fig 1. Besides the skin layer that reduces the filling rate during the injection molding process, the cooled-down polymer also contributes to the formation of a weld line, as shown in Fig 1(a). With mold temperature control assistance, the skin layer is thinner, and the plastic temperature at the weld line is higher, which could lead to better bonding, as shown in Fig 1(b). The structure of the weld line has a lower rate of continuity than the other part, therefore, it reduces the strength of the plastic parts. Many authors tried to prevent the negative effects of the weld line by adding more filler, changing the injection shape, and optimizing the injection process. Wu et al. [9], for example, investigated the influence of geometry and injection molding conditions on the strength of the weld line. They pinpointed that the most important factor that must be controlled is the melt temperature. While the packing pressure factor does not strongly affect the weld line strength. Kagitci et al. [10] indicated that a longer injection improves the flow rate in the mold, therefore, the weld line strength is enhanced. Quintana et al. [11] strengthen weld lines by using reinforcement and optimizing the injection parameters. They revealed that an optimal packing pressure leads to good fiber orientation, therefore, the weld line resistance is improved. Baradi et al. [12] also researched the weld line strength of a reinforced polymer with fiberglass. The gradual decrease in the fiber orientation gradient over the flow distance causes the rise in failure strain and failure stress at the weld line.

Polyamides (PA) polymer is typically preferred where high strength, high elasticity, high temperature resistance, and high abrasion resistance are required [13–15]. PA can be mixed with short glass fiber to produce short glass fiber-reinforced polyamide chips, which is a very conventional form for the injection molding process [16–18]. Mouhmid et al. [19] studied the impacts of strain rate, temperature, and glass fiber percentage on the mechanical properties of PA66 composite. Adding more glass fiber results in higher values of tensile strength and elastic modulus. However, the ductility of the composite suffers a decline. Increasing the molding temperature leads to a higher rate of ductility and a lower rate of stiffness. Thomason et al. [20] investigated the effects of fiber characteristics on the quality of PA6 composite. The study showed that the mechanical properties of the composite decrease dramatically in the fiberglass diameter of 10–17 micrometers. A novel approach to controlling the negative effect of the weld line is the utilization of high mold temperatures during molding, which significantly exceeds the common molding practice where temperatures are typically below 100˚C. This is facilitated by a newly designed and manufactured mold structure that enables localized

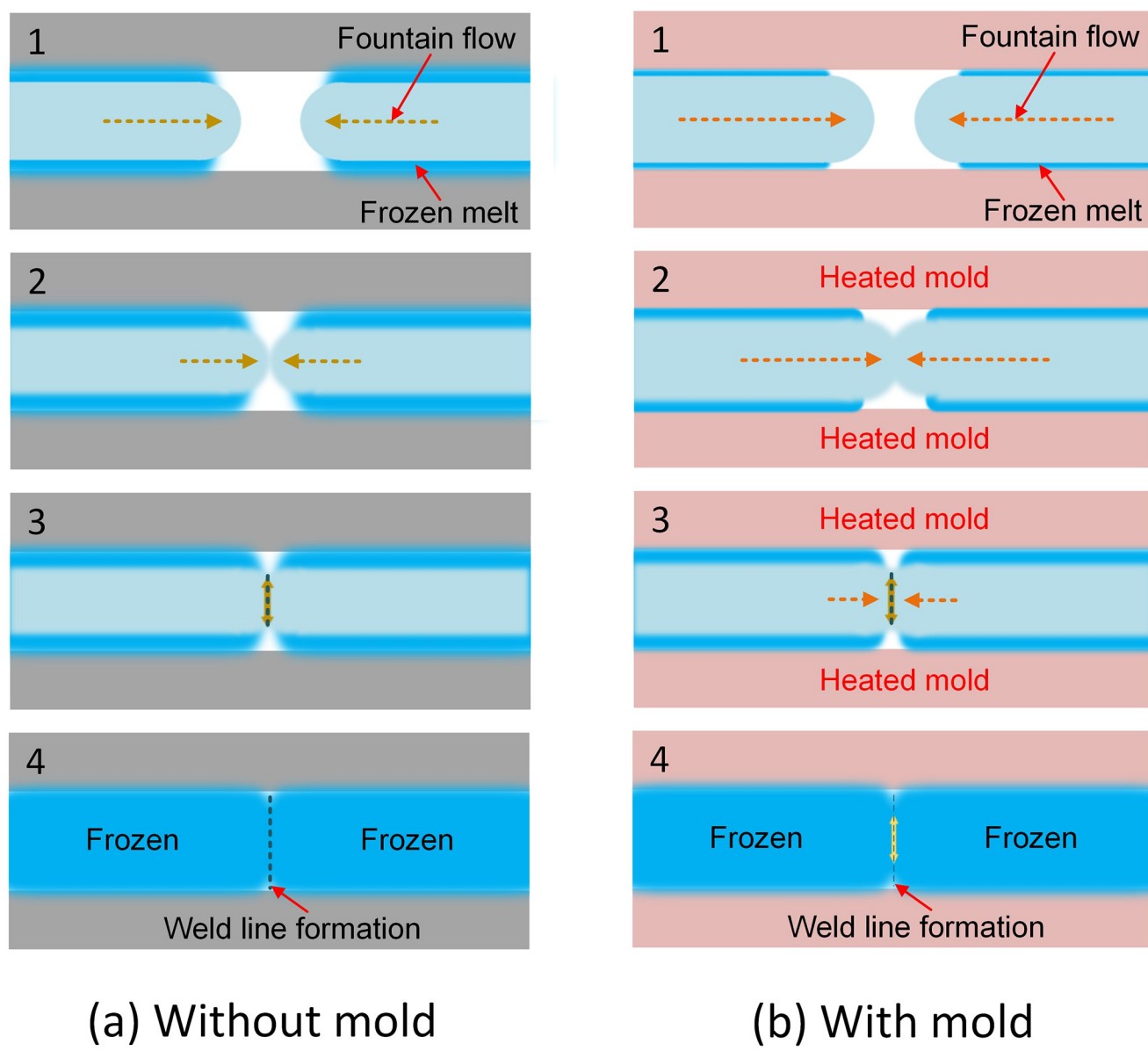

**Fig 1. The weld line formation mechanism during injection molding process: (a) without mold temperature control system, and (b) with mold temperature control system.**

heating, termed the 'novel cavity', as introduced by Meister *et al.* [21]. However, the effects of injection parameters and especially mold temperature control on the weld line strength of PA6 are rarely discussed. The optimal parameters are also not fully considered.

This study delves into the impact of injection parameters and mold temperature control on the weld line strength of composite samples made of polyamide 6 (PA6). The potential improvements in weld line strength under elevated temperature conditions could be revealed. Additionally, this paper introduces a sophisticated method of optimization by integrating an artificial neural network (ANN) with a genetic algorithm (GA). This combination allows for a

refined analysis of the molding parameters, taking the study beyond traditional experimental design and into the realm of predictive modeling and optimization. Furthermore, the research uses scanning electron microscopy (SEM) analysis to examine the fractures in the weld lines. The outcomes of this study could offer insights into enhancing the injection molding processes, where the optimization of weld line strength is paramount for the quality and durability of PA6 samples.

## 2. Experimental methods

The mold cavity shape is manufactured to form injection samples that follow ASTM D638 standards as in Fig 2. For observing the weldline strength, the part was injected with 2 molding gates, so, the two melt flow front will meet at the center of part. The injection mold was designed and manufactured with the core and cavity plates, as presented in Fig 3. This figure displays two halves of an injection mold used for plastic molding, with annotations indicating different parts and features of the mold. On the left side, labeled as the "Core plate", there is an area marked as "Heating area", which suggests this particular region of the mold is subject to controlled heating during the molding process. This could be to ensure that the material flows correctly or to maintain a certain temperature profile during injection. On the right side, labeled as the "Cavity plate", two cavities are visible: one marked as "Traditional cavity (without heating)" and the other as "Novel cavity (with local heating)". These labels indicate a comparison study between traditional mold design and a new design that includes localized heating, which is a part of the study to improve weld line appearances in molded parts. The "Weld line appearance areas" are highlighted on the traditional and novel cavity, indicating the specific locations where the weld lines typically form. Weld lines are areas where two flow fronts meet during the molding process and can be weak points in the finished product. Finally, the "Venting gaps" are marked on both sides of the cavity plate. These gaps are crucial for allowing air to escape from the mold as the material is injected, preventing defects in the final product due to trapped air. In general, this setup is likely part of the experimental method used to investigate

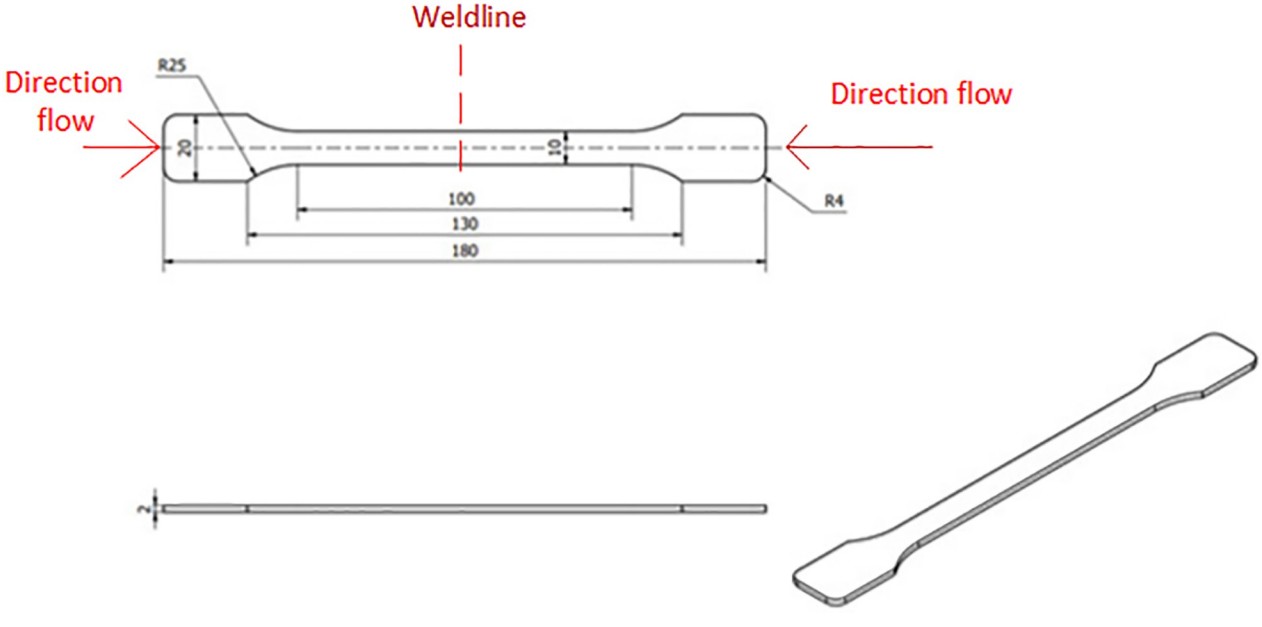

**Fig 2. Testing sample ASTM D638 (Unit: mm).**

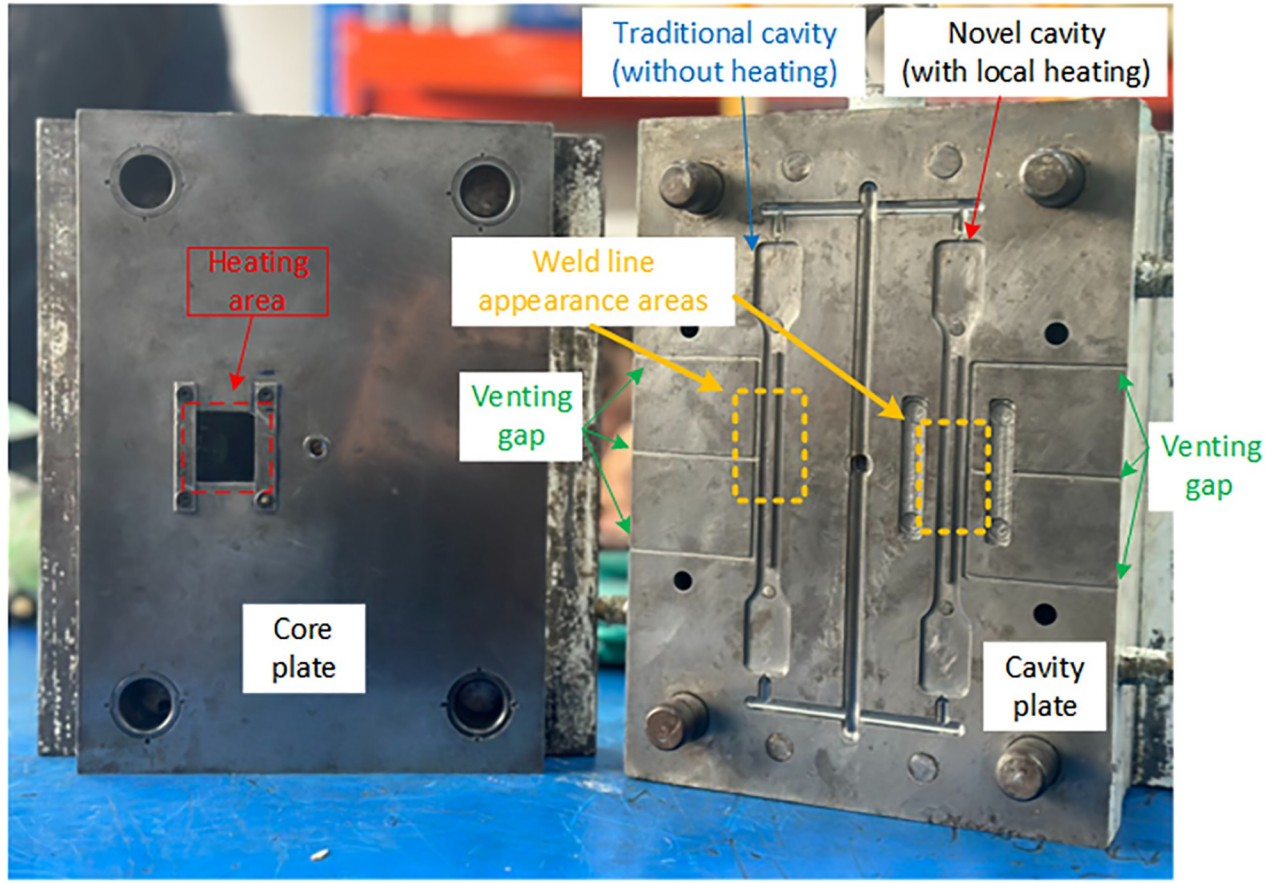

**Fig 3. Mold plates for molding process.**

the effect of mold temperature control and molding parameters on the strength of the weld lines in the molded plastic parts.

The pure PA6 material is supplied from Libolon N200, Li Peng Enterprise, Taiwan with a UTS of 85.9 MPa, an elongation value of >210%, and a melt temperature for injection molding is 240 ˚C–250 ˚C. Before injection molding, the raw materials are dried at 85 ˚C for 12 hours to eliminate moisture. During the injection molding process, the injection parameters such as filling time, packing time, melt temperature, and mold temperature is controlled as shown in Table 1. The injection molding machine used in this study is Haitian-MA 1200III, having a 3 mm nozzle with a temperature limit switch control, a 36 mm screw diameter, a 157 g injection capacity, a 122 mm/s screw speed, and a 1200 kN clamping force.

Table 1 lists various injection molding parameters for PA6 samples. It shows a systematic variation of filling time, packing time, melt temperature, and mold temperature to determine their effects on the quality of the molded polyamide 6 (PA6) parts, specifically targeting the strength of the weld lines. Before conducting the optimization process, we tried to survey the reasonable parameters that could successfully inject the sample with a good shape. Therefore, the parameter ranges are surveyed in a narrow range for better optimization results.

- **Filling Time**: This parameter is varied from 3.0 to 3.8 seconds in the initial entries (No. 1 to No. 5), and fixed at 3.4 seconds in subsequent entries. The filling time affects how quickly

**Table 1. Injection molding parameters for PA6 samples.**

| No. | Filling time (s) | Packing time (s) | Melt temperature (˚C) | Mold temperature (˚C) | UTS (MPa) | Elongation (%) |
|---|---|---|---|---|---|---|
| 1 | 3.0 | | | | 36.47 | 225.26 |
| 2 | 3.2 | | | | 39.17 | 193.34 |
| 3 | 3.4 | 0.4 | 244 | 145 | 49.93 | 394.35 |
| 4 | 3.6 | | | | 43.3 | 365.95 |
| 5 | 3.8 | | | | 45.53 | 341.97 |
| 6 | | 0 | | | 45.03 | 397.52 |
| 7 | | 0.2 | | | 37.53 | 248.52 |
| 8 | 3.4 | 0.4 | 244 | 145 | 49.93 | 394.35 |
| 9 | | 0.6 | | | 51.23 | 410.94 |
| 10 | | 0.8 | | | 55.37 | 543.96 |
| 11 | | | 240 | | 38.7 | 254.68 |
| 12 | | | 242 | | 53.5 | 532.47 |
| 13 | 3.4 | 0.4 | 244 | 145 | 49.93 | 394.35 |
| 14 | | | 246 | | 55.67 | 549.33 |
| 15 | | | 248 | | 40.8 | 307.57 |
| 16 | | | | Without heating | 45.1 | 320.5 |
| 17 | | | | 111 | 43.0 | 301.43 |
| 18 | 3.4 | 0.4 | 244 | 145 | 49.93 | 394.35 |
| 19 | | | | 153 | 47.73 | 424.09 |
| 20 | | | | 173 | 62.3 | 582.6 |

the polymer fills the mold cavity. Changes in this time can affect the quality of the weld lines where the flow fronts meet.

- **Packing Time**: Listed as varying from 0.4 seconds initially and then explored at different levels (0 to 0.8 seconds). Packing time is the duration for which pressure is applied to the molten polymer inside the mold to pack it and compensate for shrinkage. It is the extra time for extra compensation after conducting the conventional holding time.

- **Melt Temperature**: This is consistently set at 244˚C for most samples, with variations from 240˚C to 248˚C explored in some cases. The melt temperature can significantly affect the flow of the polymer and the characteristics of the weld line.

- **Mold Temperature**: Ranges from 145˚C to 173˚C, with an entry for "without heating." Mold temperature can influence the cooling rate of the polymer, which in turn affects the strength and appearance of weld lines.

Table 1 shows a controlled experiment where one parameter is varied while the others are held constant to isolate the effects of other parameters. For example, No. 1 to No. 5 vary filling time with constant packing time, melt temperature, and mold temperature. No. 6 to No. 10 keep filling and melt temperature constant while varying the packing time. No. 11 to No. 15 change the melt temperature while keeping the other parameters constant. Finally, No. 16 to No. 20 adjust the mold temperature, exploring the effects of having no heating and varying degrees of heating on the weld lines. The research utilizes gas-assisted mold temperature control to increase mold temperature from 30˚C to 173˚C, indicating a focus on the effects of mold temperature on the mechanical properties of the weld lines, which are critical for the part's strength and integrity. The average mechanical values from these samples are recorded

with error bars, suggesting statistical analysis of the results to determine the optimal parameters for the strongest weld lines. The mold was pre-heated with a Makita HG6530V heat gun before injection molding. The heating gun was adjusted to 600 ˚C hot air, 550 l/min airflow, and a distance of 10 mm. A fixture clamp was used to secure the distance. A Fluke TiS20 infrared camera (Fluke Corporation, Everett, Washington, DC, USA) was used to check the mold temperature.

In experiment, the molding cycle will be operated at least 10 cycles for reaching to the stable stage, after that, five samples will be collected. Then, the average mechanical values will be used for comparing and discussing in this paper. Tensile strength and elongation results are shown as an average with error bars. The error bars in the result diagrams represent the mechanical value's variations from an average number.

The samples are tested using an ASTM D638 standard with the tensile test equipment AG-X Plus 20 kN (Shimadzu, Japan) at a 5 mm.min$^{-1}$ speed and a grips distance of 135 mm. Following the tensile test, the fracture surfaces are studied using a scanning electron microscope (SEM) TM4000 (Hitachi, Japan).

## 3. Results and discussion

### 3.1. Effects of injection parameters

To investigate the effects of filling time, the injection molding parameters are set at 3.0–3.8 s, packing time 0.4 s, melt temperature 244 ˚C, and mold temperature 145 ˚C. Fig 4 presents the stress-strain diagrams of sample 1. The effects of filling time on the tensile test results are shown in Fig 5. Typically, the stress-strain diagram of the pure PA6 polymer has a smoother curve as there is no fiberglass in the polymer matrix [15]. Interestingly, the diagram has a rough shape, indicating the non-uniform behavior of the weld line structure during the tensile test process. Fig 3 compares the ultimate tensile strength (UTS) values of PA6 composite samples with weld line at various filling times. The UTS values are 36.47 MPa, 39.17 MPa, 49.93 MPa, 43.3 MPa, and 45.53 MPa corresponding to the filling times of 3.0 s, 3.2 s, 3.4 s, 3.6 s, and 3.8 s. Compared to the UTS value of the composite without the existence of the weld line, which is 80 MPa, the UTS values of all samples obtain a lower rate. This reduction is due to the discontinuity of the weld line structure [22, 23]. Moreover, applying the filling time that is higher than 3.0 s results in an improvement in the UTS. A Filling time that is too short may require higher pressure and speed, reducing the smoothness of the injection molding process due to the effect of the viscous flow mechanism [24]. Excessive filling time, on the other hand, might result in a higher rate of solidification during the injection process. Therefore, at 3.6 s and 3.8 s, the UTS value is lower than in the case of 3.4 s due to the excessive filling time. The optimal UTS value of 49.93 MPa is achieved at a filling time of 3.4 s.

The comparison of the elongation values of PA6 samples with weld line at various filling times is shown in Fig 6. The elongation values are 225.26%, 193.34%, 394.35%, 365.95%, and 341.97%, which correspond to the filling times of 3.0 s, 3.2 s, 3.4 s, 3.6 s, and 3.8 s. The result shows that the filling time has a strong impact on the elongation value of the PA6 samples. In addition, samples with the filling time of 3.4 s, 3.6 s, and 3.8 s have a greatly higher elongation value than samples with the filling time of 3.0 s, and 3.2 s. The highest elongation of 394% is gained at the filling time of 3.4 s. Remarkably, at 3.4 s, the PA6 sample achieves both the highest UTS value of 49.93 MPa and the highest elongation value of 394.35%. It means that at 3.4 s, the PA6 sample achieves the balance between the UTS value and the elongation value.

The packing step is employed to completely compress the plastic as it cools and shrinks in the mold. Using a packing step during the injection process could improve sample quality by eliminating air bubbles [25]. This study examines the effects of packing times ranging from 0

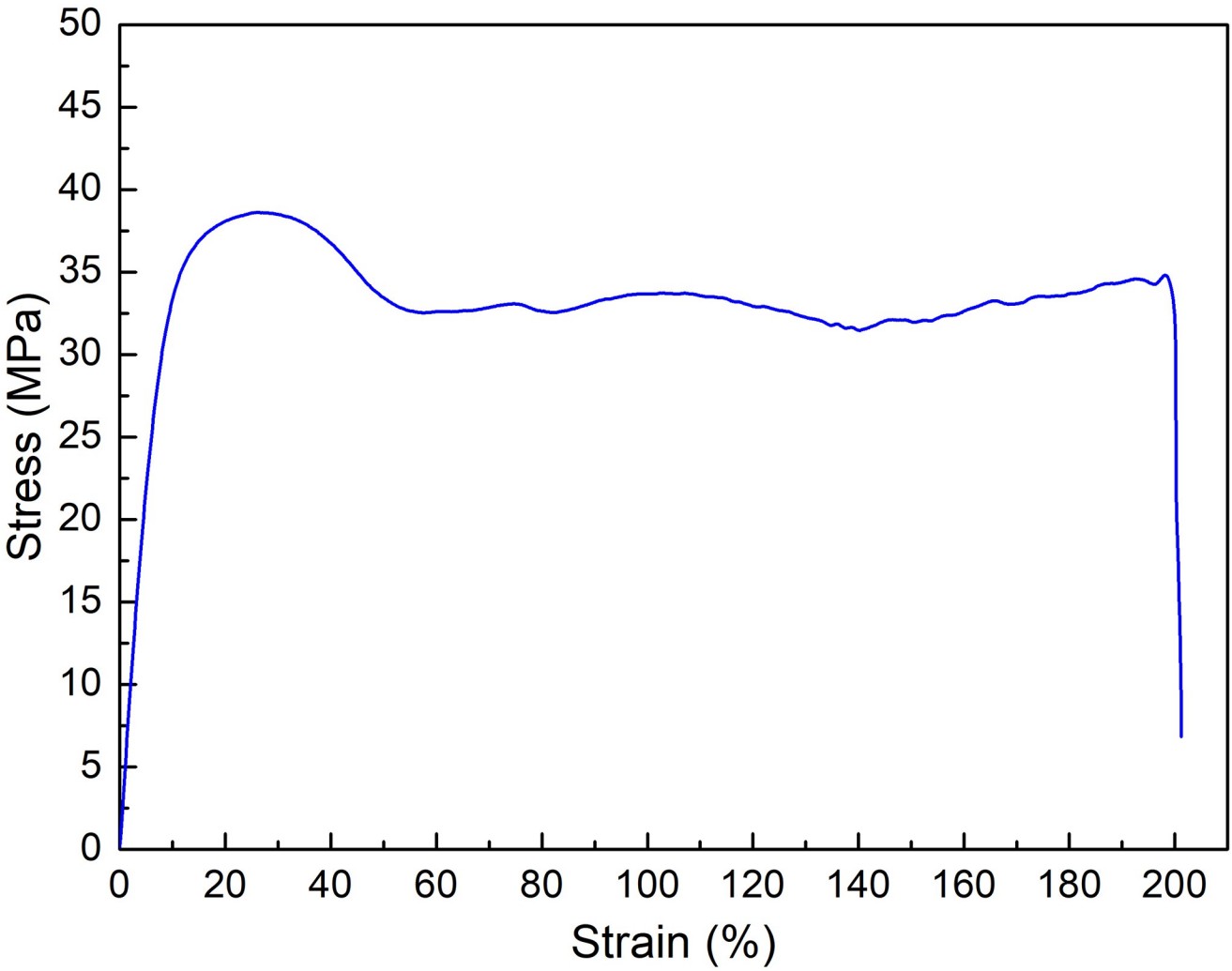

**Fig 4. Stress-strain diagrams of sample 1 of PA6 composite.**

to 0.8 seconds, filling time 3.4 s, melt temperature 244 ˚C, and mold temperature 145 ˚C, as shown in Table 1. Fig 7 displays the UTS values of PA6 samples with weld line at different packing times. The UTS values are 45.03 MPa, 37.53 MPa, 49.93 MPa, 51.23 MPa, and 55.37 MPa corresponding to 0 s, 0.2 s, 0.4 s, 0.6 s, and 0.8 s. Packing time at 0.2 s seems to not enough for an improvement in the UTS value. However, from 0.4 s to 0.8 s packing time, the UTS value of the PA6 sample experiences an increase compared to sample without packing step. In general, 0.8 s is the best packing time for the greatest UTS value of 55.37 MPa.

The comparison of the elongation values of PA6 samples with weld line at various packing times is shown in Fig 8. The elongation values are 397.52%, 248.52%, 394.35%, 410.94%, and 543.96%, corresponding to 0 s, 0.2 s, 0.4 s, 0.6 s, and 0.8 s. The changing pattern of the elongation result is similar to the UTS value, as shown in Fig 7. Compared to a sample without packing, which is 0 s case, applying a packing time of 0.2 s seems too short for an improvement in the elongation value. From 0.4 s to 0.8 s, the elongation values are greater than the sample without packing. At 0.8 s, the elongation value of the PA6 sample with weld line reaches the highest value of 543.96%. It is noteworthy that at 0.8 s, the sample achieves the best tensile

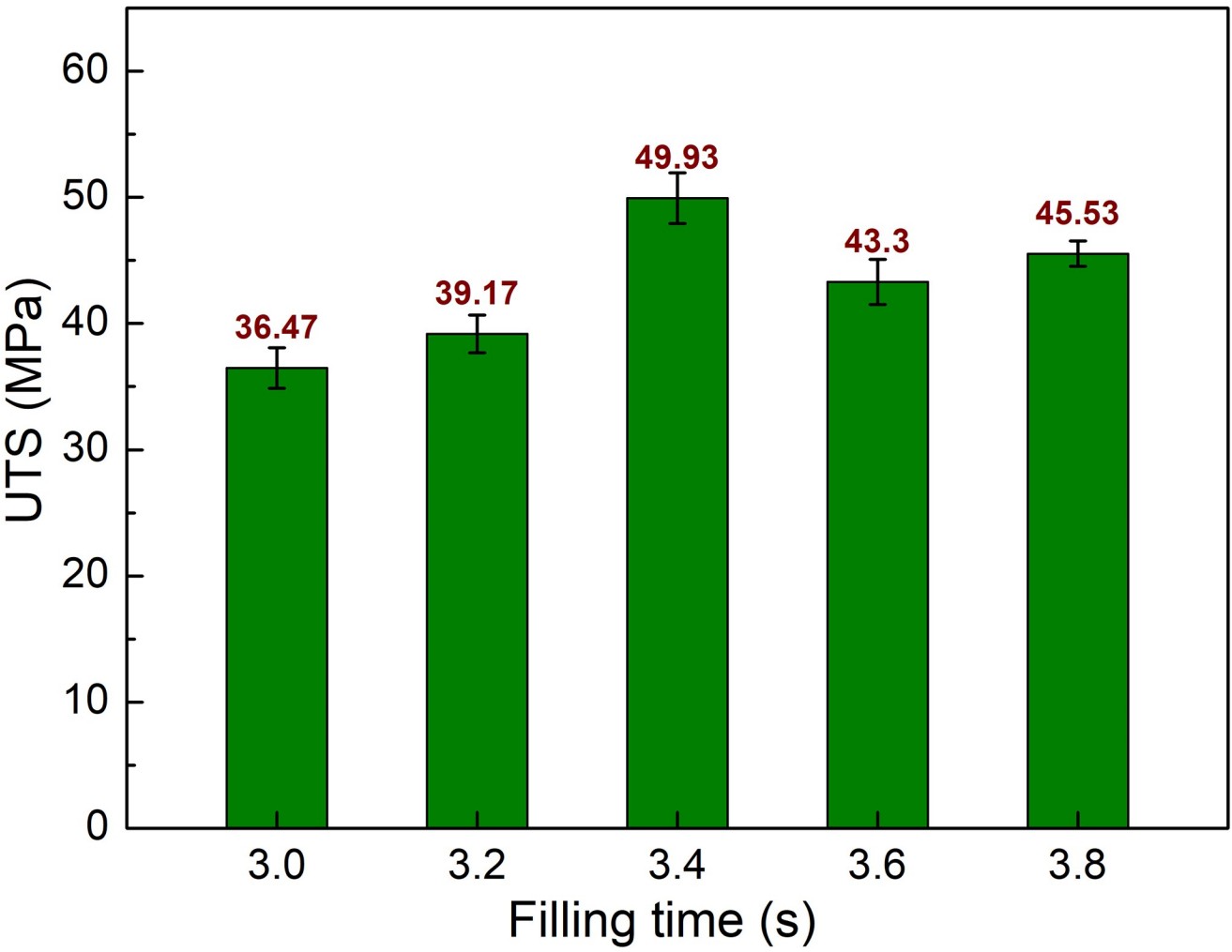

**Fig 5. Comparison of UTS values of PA6 samples with weld line at different filling times.**

strength with the highest UTS value and the highest elongation value. The explanations could be the removing of the air bubbles [25].

Besides filling time and packing time, the melt temperature of the PA6 is also investigated in this study. Melt temperature is an important aspect that has a significant impact on the properties of plastics. High melt temperatures might cause polymer degradation and lower the properties of plastics. This study examines the influences of melt temperature ranging from 240 ˚C to 248 ˚C, filling time of 3.4 s, packing time of 0.4 s, and mold temperature of 145 ˚C, as shown in Table 1. Fig 9 presents the comparison of UTS values of PA6 samples with weld lines at different melt temperatures. The UTS values are 38.7 MPa, 53.5 MPa, 49.93 MPa, 55.67 MPa, and 40.8 MPa corresponding to 240 ˚C, 242 ˚C, 244 ˚C, 246 ˚C, and 248 ˚C. From 242 ˚C to 246 ˚C, the UTS value increases slightly, the UTS value of the sample is higher than cases of 240 ˚C and 248˚C. Because the low melt temperature could increase the viscosity, therefore, the filling rate is reduced [26]. The highest UTS value is achieved at 246 ˚C, while the higher temperature cases suffer a decline. The reason for the reduction could be the polymer degradation at high temperatures [27].

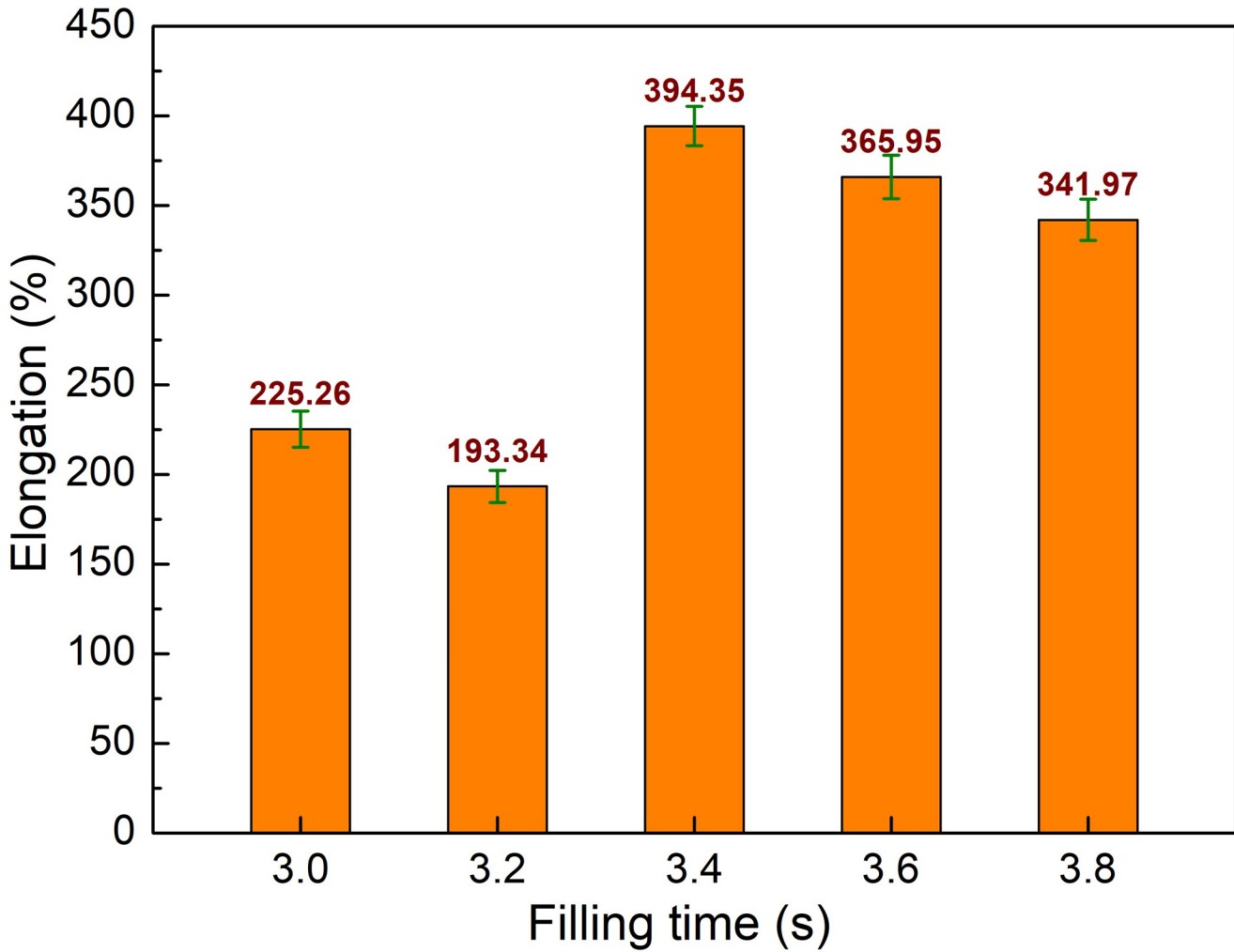

**Fig 6. Comparison of elongation values of PA6 samples with weld line at different filling times.**

The elongation values of PA6 samples with weld lines at different melt temperatures are shown in Fig 8. The elongation values are 254.68%, 532.47%, 394.35%, 549.33%, and 12.27% corresponding to 240 ˚C, 242 ˚C, 244 ˚C, 246 ˚C, and 248 ˚C. Notably, the changing pattern of the elongation value, as shown in Fig 10, is similar to the UTS, as shown in Fig 9. At 246 ˚C, the sample obtains the highest elongation value of 549.33%. However, at 240 ˚C, the elongation value is only 254.68%, which is the lowest value. The reason for these phenomenon are too high melt temperature could cause polymer degradation, while too low melt temperature can lead to poor filling rate [27, 28]. Overall, at 246 ˚C, the PA6 samples with weld line have both good UTS and elongation values.

The detailed analysis of Figs 11 and 12 provides a clear depiction of the correlation between mold temperature and the mechanical properties of PA6 samples, specifically ultimate tensile strength (UTS) and elongation. The data indicates that an increase in mold temperature correlates with a significant improvement in the mechanical performance of the weld lines within the samples. Fig 11 illustrates that the UTS values of PA6 samples with weld lines ascend with increasing mold temperatures. At room temperature, the UTS starts at 45.1 MPa and shows a

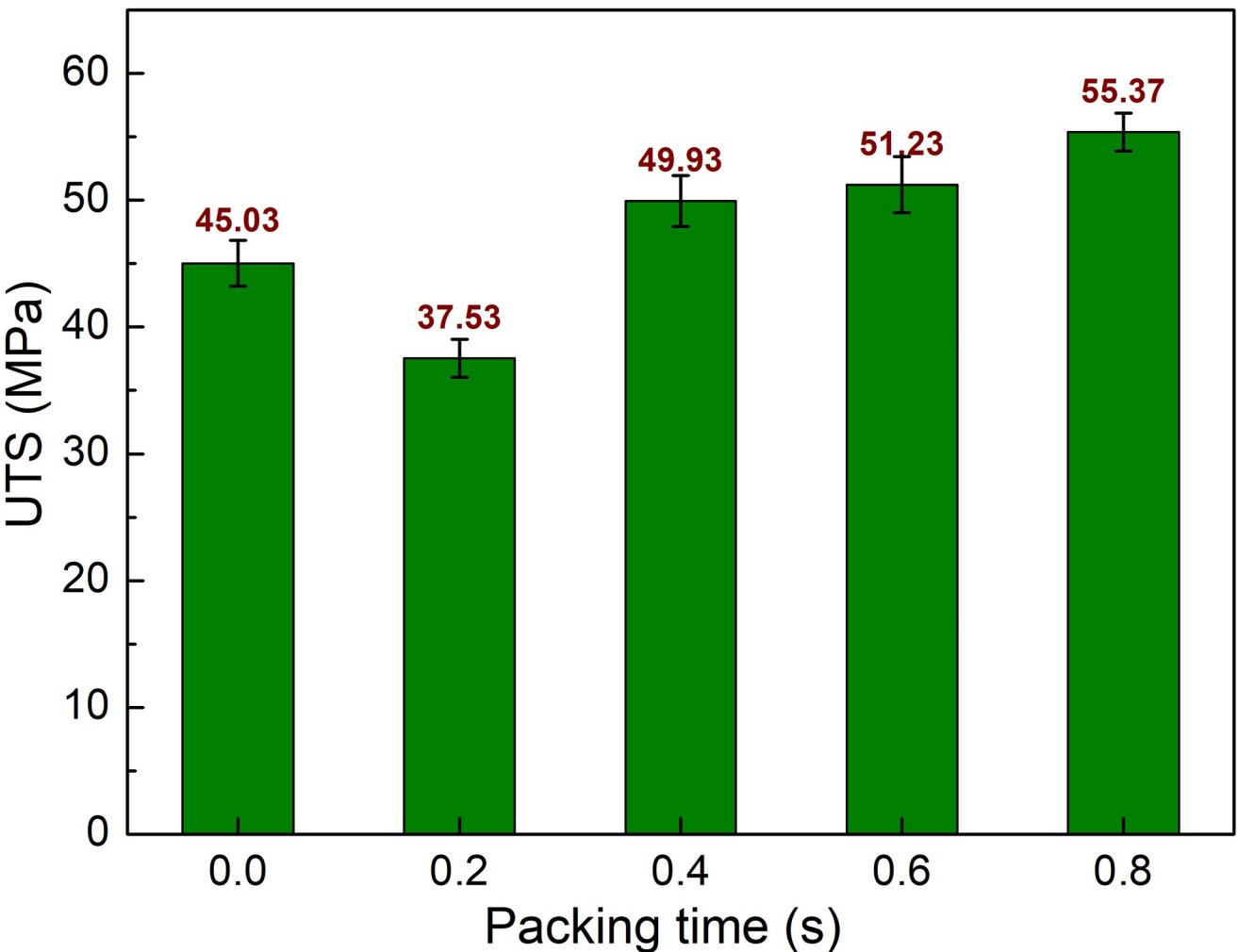

**Fig 7. Comparison of UTS values of PA6 samples with weld line at different packing time.**

variable trend, with a slight decrease at 111 ˚C (43 MPa), followed by an increase at 145 ˚C (49.93 MPa) and a subsequent minor drop at 153 ˚C (47.73 MPa). However, a substantial increase to 62.3 MPa is observed at the highest tested temperature of 173 ˚C. This jump in UTS at the highest mold temperature suggests a markedly enhanced fusion of the polymer chains at the weld lines, indicating a stronger bond and greater overall structural integrity of the molded product.

Similarly, Fig 12 displays the elongation percentage of the samples, which also exhibits a dramatic increase as the mold temperature rises, with the most significant improvement noted at 173 ˚C, reaching an elongation value of 582.6%. This implies not only stronger but also more flexible weld lines at higher mold temperatures, which can absorb more energy before failure, a desirable property in many applications where flexibility and durability are required. The discussion of these results underscores the benefit of increasing mold temperature on the weld line strength in PA6 samples. By optimizing the mold temperature, the injection molding process can produce components with superior mechanical properties, such as increased strength and flexibility, which are critical factors in many industrial applications. The

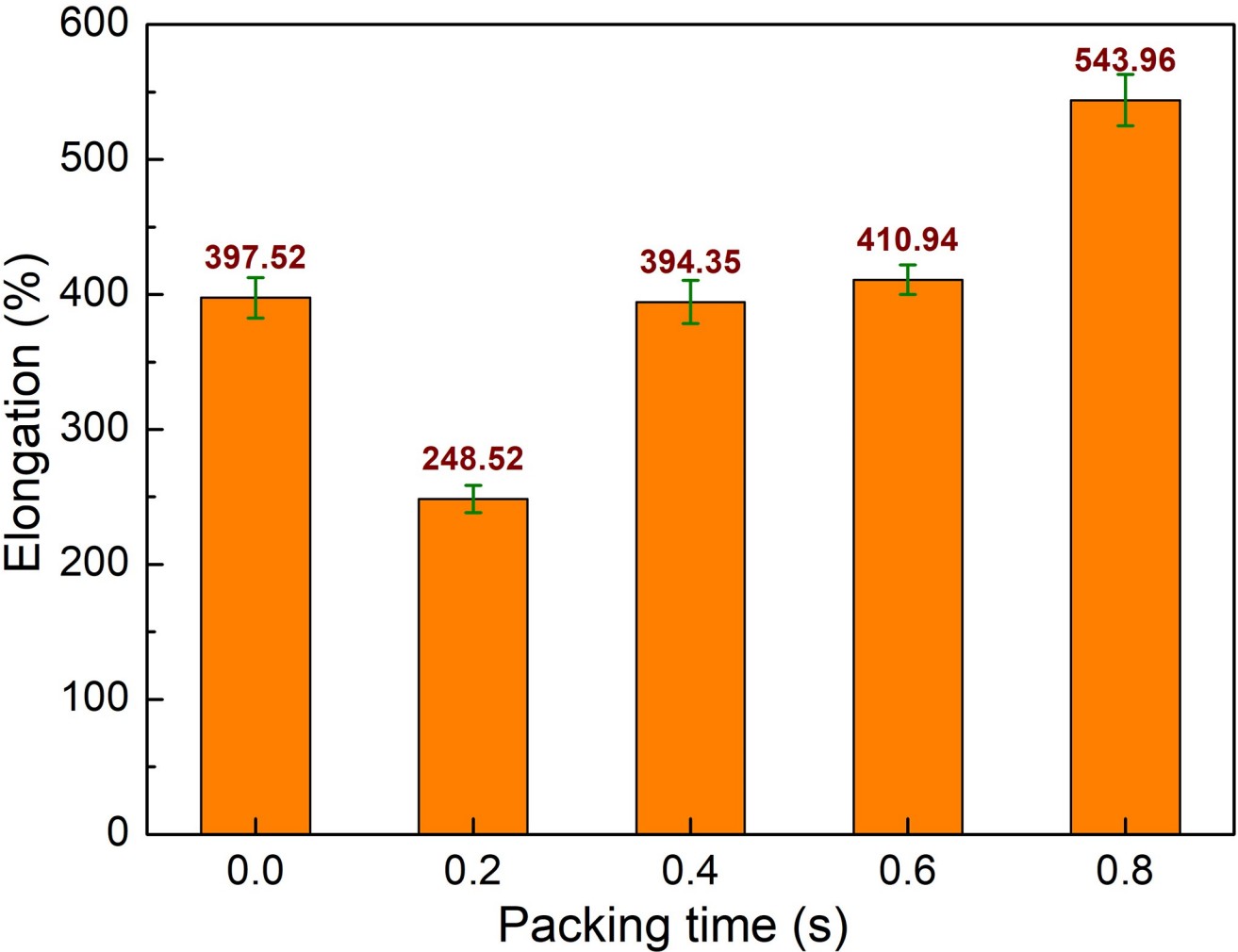

**Fig 8. Comparison of elongation values of PA6 samples with weld line at different packing time.**

enhanced performance at higher mold temperatures can lead to products with longer life spans and better resistance to mechanical stress, which is particularly beneficial in automotive, aerospace, and consumer goods industries where the reliability and durability of plastic components are of paramount importance.

In practice, this means that the injection molding industry can achieve higher quality in their products with the adoption of high-temperature molds. This not only elevates the product standards but can also reduce failure rates and warranty claims, leading to improved customer satisfaction and potentially lower manufacturing costs due to fewer rejected parts. Moreover, the ability to produce stronger and more resilient parts may open new avenues for the use of PA6 in applications where it was previously not considered suitable, thus expanding the material's market potential.

In summary, the parameters for a good UTS value that is higher than 50 MPa corresponds to samples 9, 10, 12, 14, and 20, as shown in Table 2. While samples with a good elongation value that is higher than 500% are samples 10, 12, 14, and 20. Therefore, samples 10, 12, 14, and 20 achieve both high UTS values and elongation values. Among them, the optimal

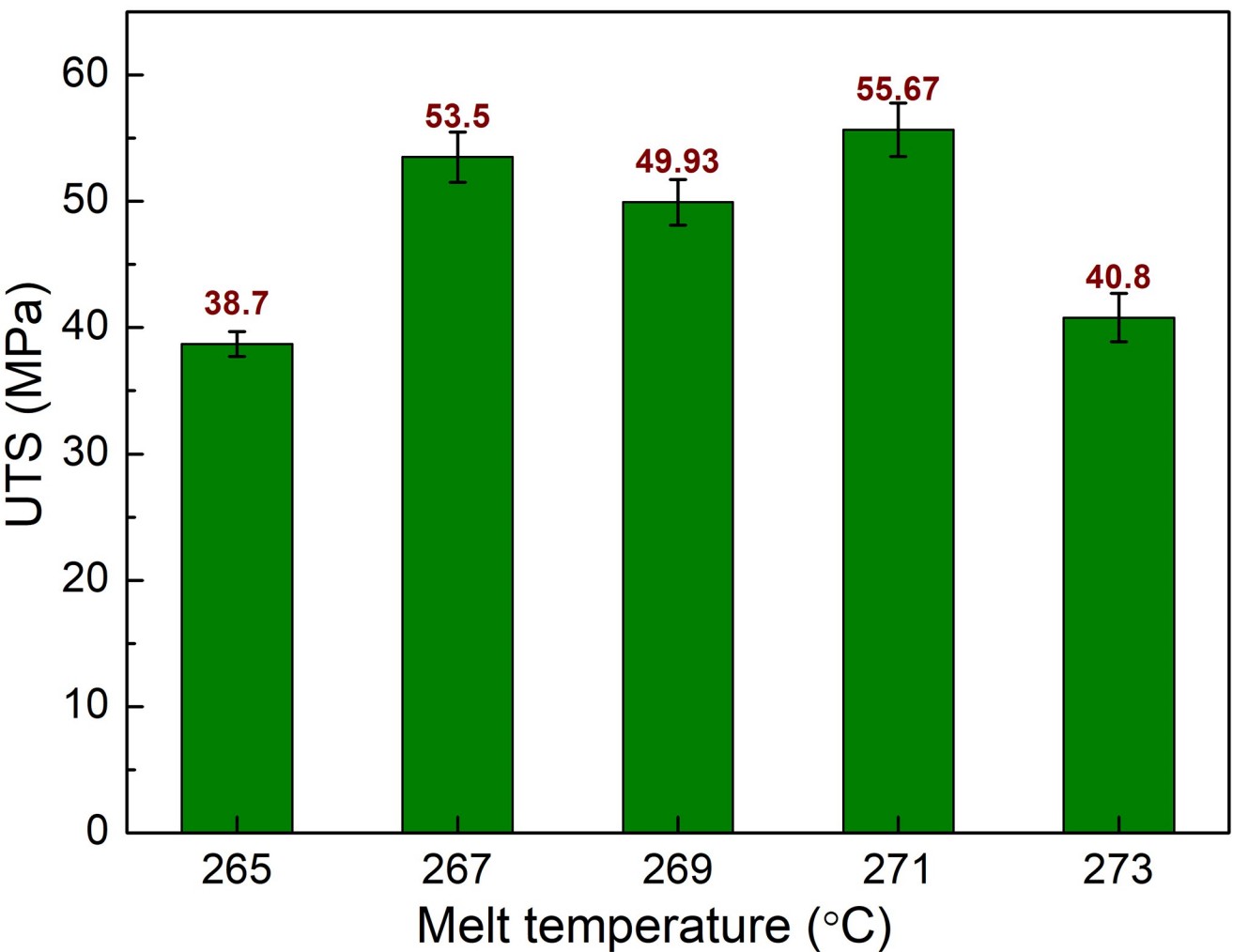

**Fig 9. Comparison of UTS values of PA6 samples with weld line at different melt temperatures.**

molding parameters of sample 20 are a filling time of 3.4 s, a packing time is 0.4 s, a melt temperature of 244 °C, and a mold temperature of 173 °C.

## 3.2. Optimization of molding parameters

The study employed a dataset of 20 samples, encompassing experiments with both PA6+30% GF and PA6+0%GF materials. To ensure robust model development and evaluation, this data underwent a random split. Following a common practice in machine learning, the split allocated 70% (14 samples) for training the model. This training data provides the foundation for the model to learn the underlying relationships between input variables and desired outputs. The remaining 30% of the data was further divided into validation (15%, 3 samples) and testing sets (15%, 3 samples). The validation set serves as a crucial intermediary step. It allows the model to be fine-tuned and identify potential overfitting issues before final evaluation. Finally, the testing set (15%, 3 samples), unseen by the model during training and validation, provides an unbiased assessment of the model's generalizability and performance on new, unseen data. Notably, to prevent bias and ensure data diversity within each set, samples from both PA6

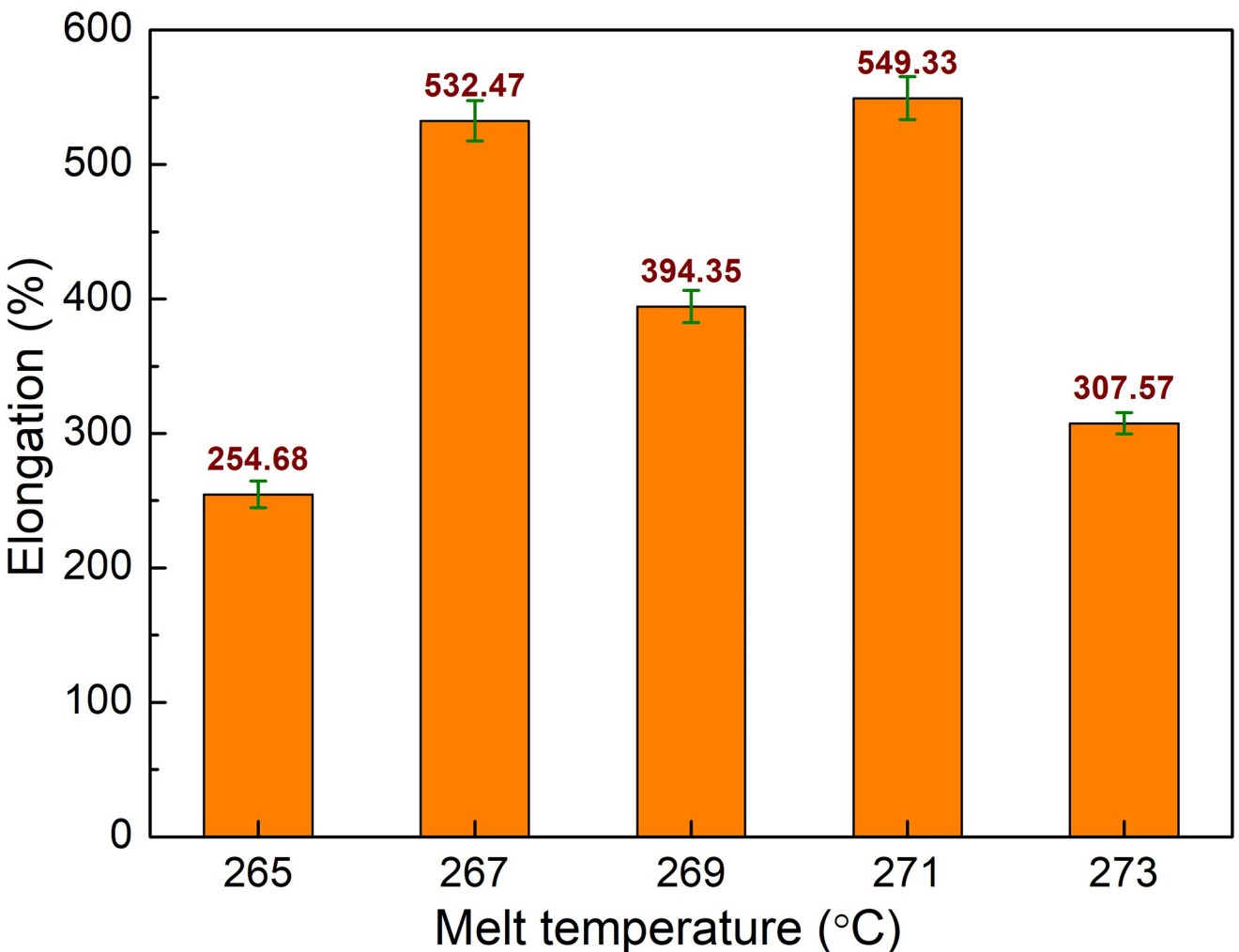

**Fig 10. Comparison of elongation values of PA6 samples with weld line at different melt temperatures.**

+30%GF and PA6+0%GF experiments were included in all training, validation, and testing sets. This approach helps the model learn complex relationships across a broader range of scenarios, ultimately leading to a more robust and generalizable model.

In constructing the Artificial Neural Network (ANN) model, a 75/15/15 split was implemented on the experimental data to create distinct training, validation, and testing sets. This ensures the model is trained on a representative portion of the data (75%), its performance is evaluated on unseen data during validation (15%), and its generalizability to entirely new data is assessed on the testing set (15%). The chosen architecture employs a single hidden layer comprised of 10 neurons. Hidden layers act as intermediate processors within the network, allowing it to learn complex relationships between input features and desired outputs. The Levenberg-Marquardt algorithm was selected for training the model. This optimization algorithm is known for its efficiency and effectiveness in handling non-linear relationships often present in real-world data. Finally, the training process was executed for 8 epochs. Epochs represent complete cycles where the entire training dataset is passed through the network once. The chosen number of epochs suggests the model achieved a satisfactory level of learning within this timeframe.

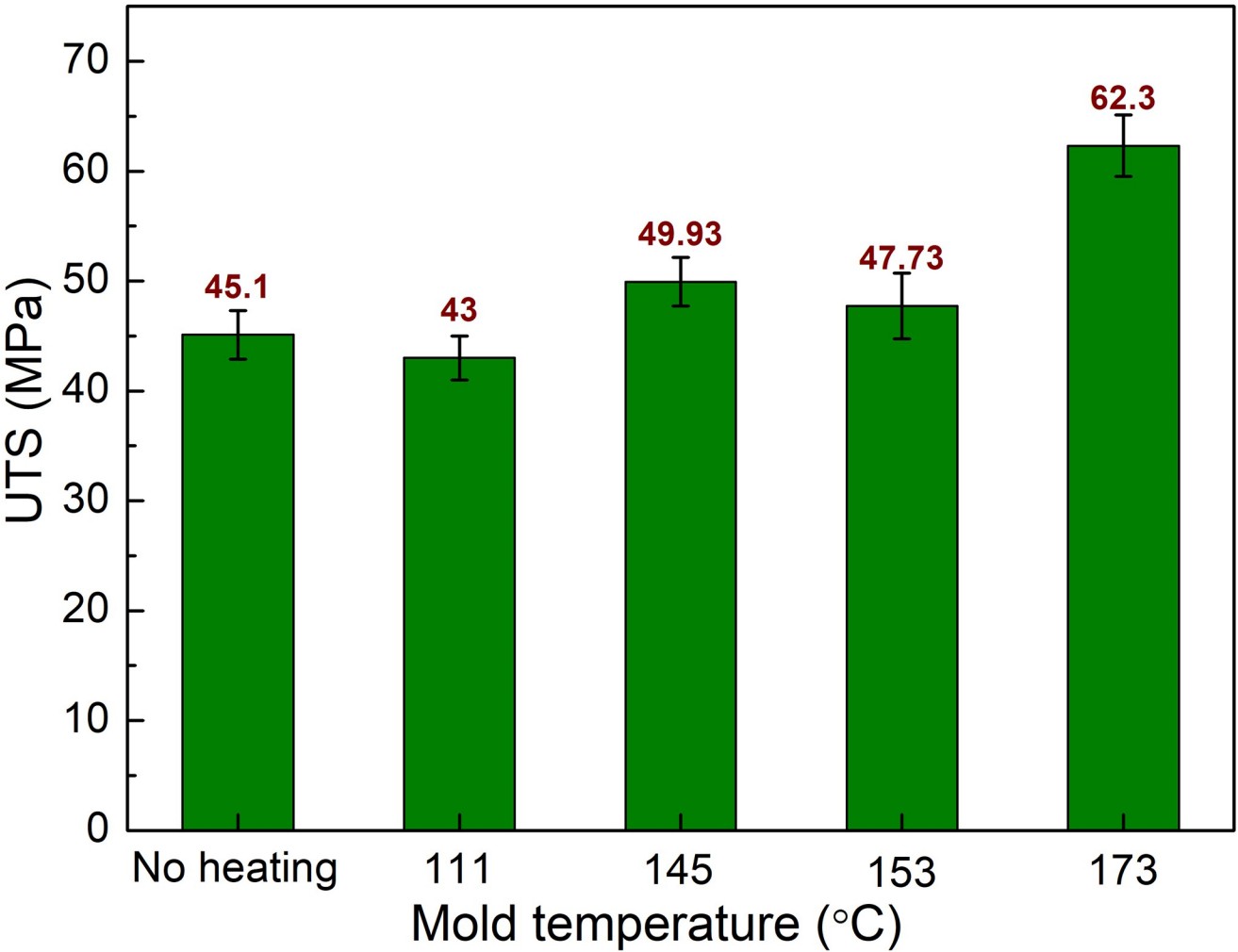

**Fig 11. Comparison of UTS values of PA6 samples with weld line at different mold temperatures.**

The genetic algorithm employed a user-defined fitness function (details not provided) to evaluate potential solutions. Each solution, represented by a chromosome, had its variables constrained within specific bounds (also not specified). Selection for breeding the next generation used a Stochastic Uniform approach, where individuals with higher fitness have a greater chance of being chosen. Mating pairs then underwent Intermediate crossover, where genes from each parent are averaged to create offspring. To maintain feasibility, Constraint-dependent mutation was applied, modifying only genes that violated problem constraints. The specific stopping criteria used to terminate the algorithm (e.g., maximum generations, convergence threshold) are not mentioned and require further investigation.

Besides the conventional comparison, this study applied the ANN method to reveal more insight into the optimized parameters, using Matlab R2014a. First, the neural network is trained with the input data and output, which are injection parameters and the UTS value of the samples. After training, the R-squared results of PA6 are present in Fig 13. The correlation between variables in a regression model is measured by R-squared. It denotes the degree of agreement between the expected and actual values. Fig 13 shows that R approaches 1,

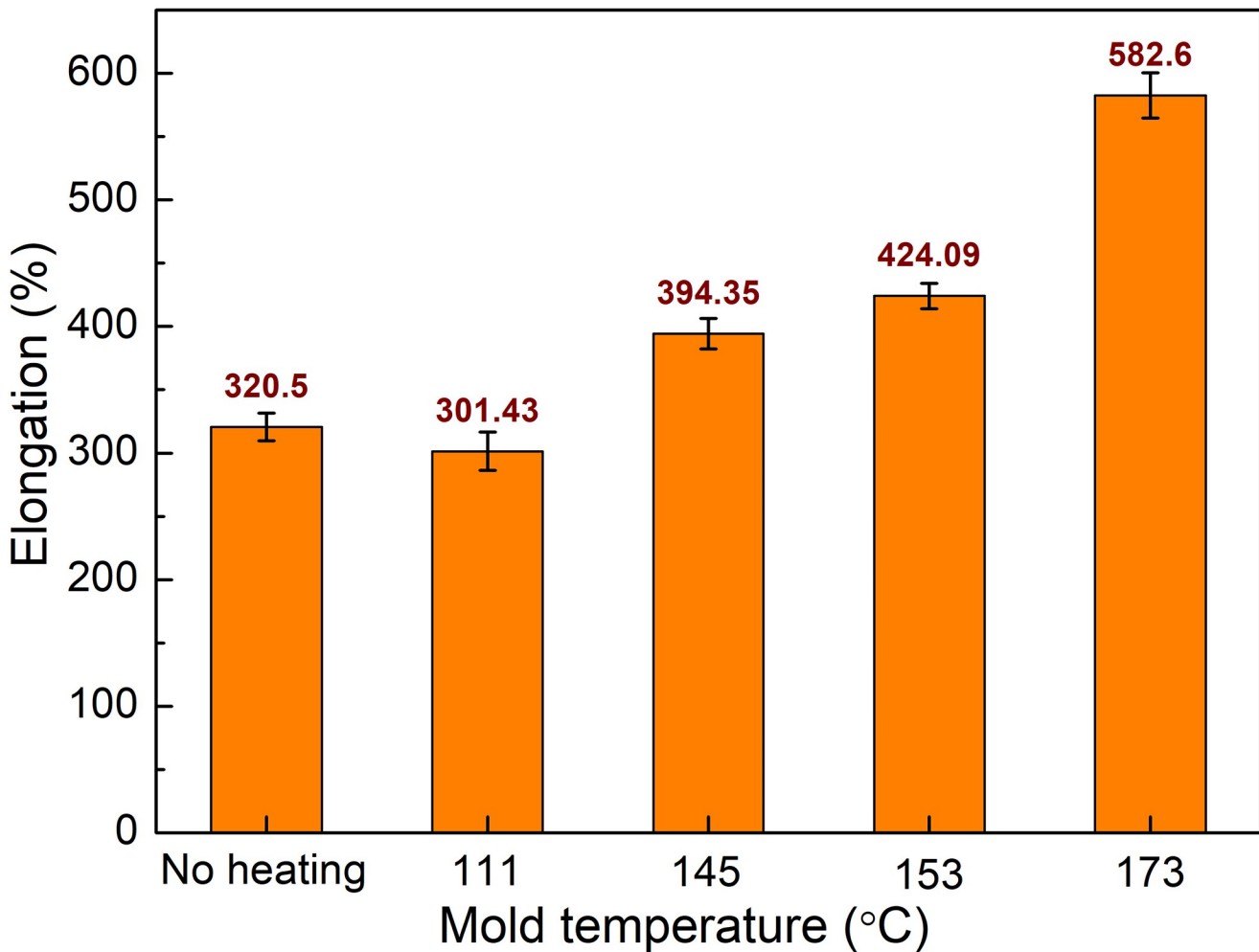

**Fig 12. Comparison of elongation values of PA6 samples with weld line at different mold temperatures.**

indicating a significant positive correlation between the variables. This suggests that the model predicts well and can explain a significant portion of the variance in the actual data.

GA (Genetic Algorithm) is used to optimize the results after training the neural network for the input and output data of PA6. The goal is to determine the best parameters that produce the best results. Using the gamultiobj algorithm tool to perform GA for 4 input variables corresponding to 4 input parameters: filling time, packing time, melt temperature, and mold

**Table 2. Summary of molding parameters for good UTS value and elongation value of PA6 samples.**

| No. | Filling time (s) | Packing time | Melt temperature | Mold temperature | UTS (MPa) | Elongation (%) |
| --- | --- | --- | --- | --- | --- | --- |
| | | (s) | (°C) | (°C) | | |
| 9 | 3.0 | 0.6 | 244 | 145 | 51.23 | 410.94 |
| 10 | 3.2 | 0.4 | 244 | 145 | 55.37 | 543.96 |
| 12 | 3.4 | 0.4 | 242 | 145 | 53.50 | 532.47 |
| 14 | 3.4 | 0.4 | 246 | 145 | 55.67 | 549.33 |
| 20 | 3.4 | 0.4 | 244 | 173 | 62.30 | 582.60 |

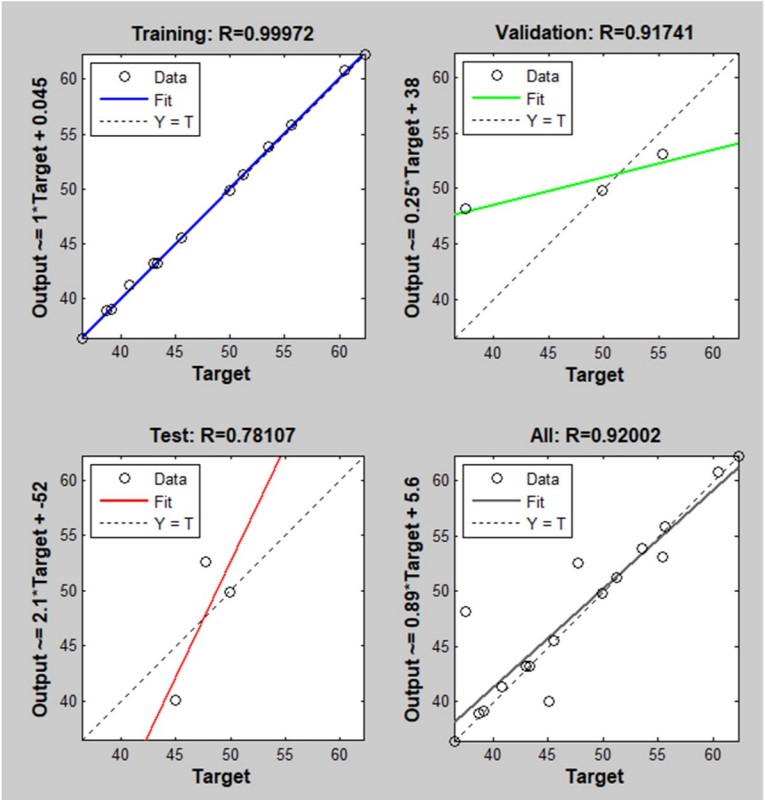

**Fig 13. The R-squared results of PA6 of the ANN method.**

temperature. The lower bound is [3 0 240 3]. The upper bound is [3.8 0.8 248 173]. In conclusion, the optimal molding parameters generated from ANN and GA methods are a filling time of 3.8 s, a packing time is 0.8 s, a melt temperature of 244 ˚C, and a mold temperature of 35.2 ˚C, as shown in Table 3.

Finally, the indication of which factors play the most important role is surveyed. The ANOVA analysis is applied via Minitab software. The response table for means of the PA6 samples with weld lines of the UTS and elongation values is shown in Tables 4 and 5. With the UTS value, the mold temperature factor has the highest rank, followed by the packing time and melt temperature factors, while the filling time has the lowest rank. Considering the elongation at break value, the packing time factor is the most important factor, followed by the melt temperature and mold temperature factors, while the filling time also has the lowest rank.

This paper focuses on the method for mold temperature control, as well as the way to apply the ANN and GA to optimize the parameters. The study just shows the optimization by the software. In the next research, the optimization will be applied in detail and the comparison between prediction and real experiment will be achieved.

Fig 14 illustrates the fracture surfaces of the PA6 samples with weld line at various magnifications. The SEM result shows the ductile fracture of the PA6 samples with the rough surface.

**Table 3. The table of optimized results for PA6.**

| PA6 | Filling time (s) | Packing time (s) | Melt temperature (˚C) | Mold temperature (˚C) |
|---|---|---|---|---|
| ANN + GA | 3.8 | 0.8 | 244 | 35.2 |

**Table 4. Response table for means of UTS values.**

| Level | Filling time | Packing time | Melt temperature | Mold temperature |
|---|---|---|---|---|
| 1 | 36.47 | 45.03 | 38.70 | 45.10 |
| 2 | 39.17 | 37.53 | 53.50 | 43.00 |
| 3 | 48.15 | 46.25 | 46.28 | 45.56 |
| 4 | 43.30 | 51.23 | 55.67 | 47.73 |
| 5 | 45.53 | 55.37 | 40.80 | 62.30 |
| Delta | 11.68 | 17.84 | 16.97 | 19.30 |
| Rank | 4 | 2 | 3 | 1 |

**Table 5. Response table for means of elongation values.**

| Level | Filling time | Packing time | Melt temperature | Mold temperature |
|---|---|---|---|---|
| 1 | 225.3 | 397.5 | 254.7 | 320.5 |
| 2 | 193.3 | 248.5 | 532.5 | 301.4 |
| 3 | 405.2 | 368.7 | 365.4 | 366.6 |
| 4 | 365.9 | 410.9 | 549.3 | 424.1 |
| 5 | 342.0 | 544.0 | 307.6 | 582.6 |
| Delta | 211.9 | 295.4 | 294.7 | 281.2 |
| Rank | 4 | 1 | 2 | 3 |

The ductile fracture mechanism of the PA6 samples is the result of low stiffness, high elongation, and high energy absorption of the polymer [29, 30]. Moreover, there are some grooves on the surface, indicating the poor bonding structure of the weld line area. Interestingly, in sample 7, which has a packing time of 0.2 s, there is a void in the microstructure of the matrix, as shown in Fig 14(b2). The void causes a reduction in the polymer quality. Generally, applying extra packing time helps remove air bubbles. However, too short a packing time could cause an improper result due to the elastic deformation rather than the plastic deformation of the polymer sample after releasing the packing step. Therefore, the air bubble could be introduced again after releasing the packing step. This is the reason for the low UTS and elongation values of this sample, as shown in Figs 7 and 8. A packing time that is higher than 0.2s leads to a positive effect as the UTS and elongation values are improved. Fig 14(d1)–14(d2) shows the fracture surface of sample 16, which is the sample without heating. Compared to the other samples with mold heating, as shown in Fig 14(a1)–14(c1), the surface has a little rougher surface, mostly indicating a lower level of bonding.

## 4. Conclusion

This study investigates the impact of injection parameters on the weld line strength of the PA6 samples. The effects of filling time, packing time, melt temperature and mold temperature on the characteristics of the weld line are examined. The parameters results are optimized by considering each parameter, each sample, and the ANN and GA methods. Some important results might be withdrawn, including:

- When considering each parameter, the optimal UTS and elongation values are achieved at a filling time of 3.4 s, a packing time of 0.8 s, a melt temperature of 246 ˚C, and a mold temperature of 173 ˚C.

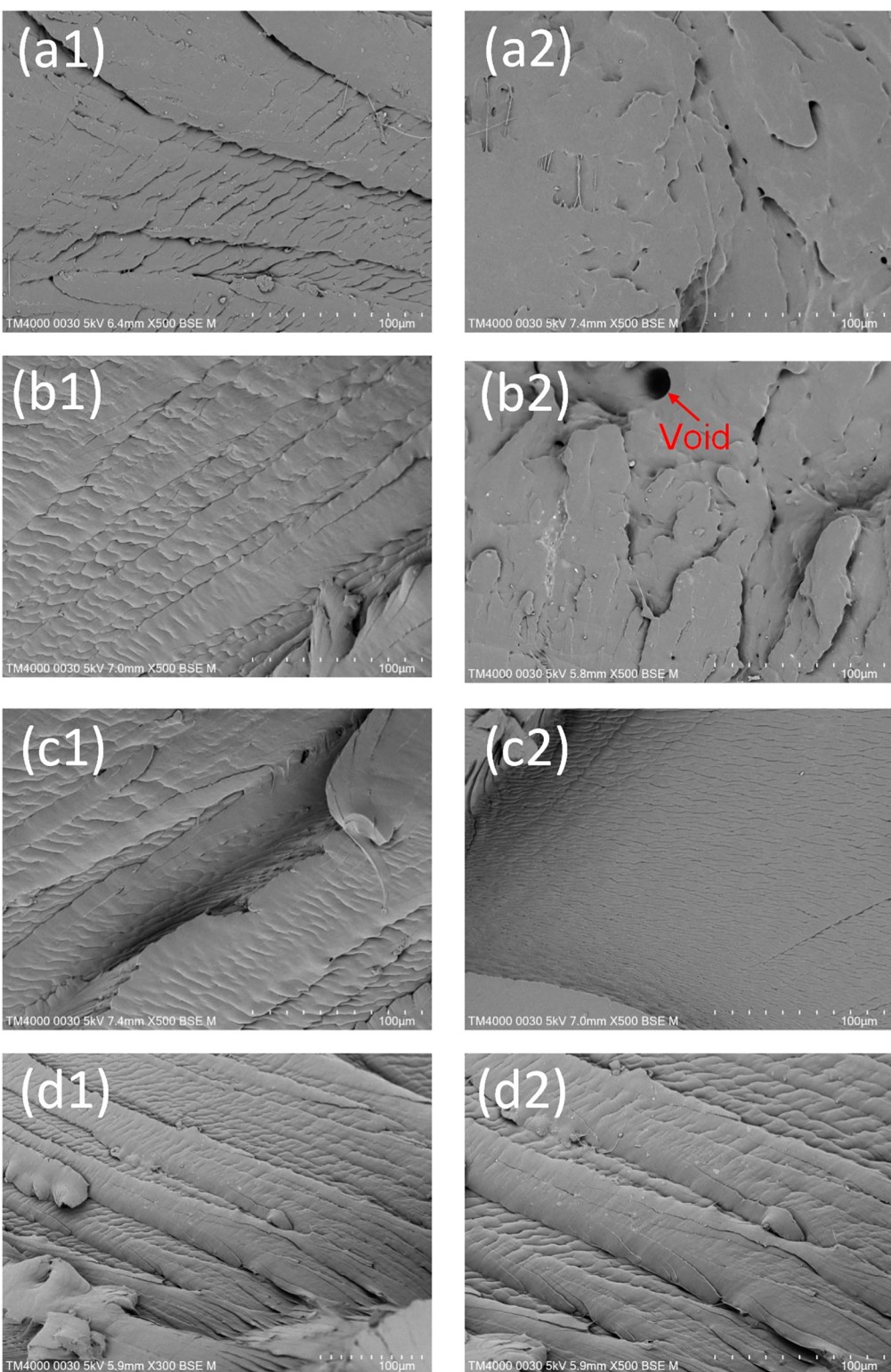

**Fig 14. SEM picture of fracture surface of PA6 samples with weld line: (a1)–(a2) sample 3, (b1)–(b2) sample 7, (c1)–(c2) sample 12, and (d1)–(d2) sample 16.**

- The optimal molding parameters of the best sample that have been conducted are a filling time of 3.4 s, a packing time is 0.4 s, a melt temperature of 244 ˚C, and a mold temperature of 173 ˚C. This parameter products sample with the UTS value of 62.3 MPa and the elongation value of 582.6%.

- The optimal molding parameters calculated from ANN and GA methods are a filling time of 3.8 s, a packing time is 0.8 s, a melt temperature of 244 ˚C, and a mold temperature of 35.2 ˚C.

- The melt temperature factor has the strongest impact on the UTS and elongation values of the PA sample with weld line. In reverse, the filling time factor has the weakest impact rate.

- The SEM image reveals a ductile fracture of the PA6 sample with a rough surface. Furthermore, there are some grooves on the surface, indicating poor bonding of the weld line area. The findings could help to improve injection molding conditions, resulting in better weld lines in the injected sample.

## Author Contributions

**Conceptualization:** Pham Son Minh, Van-Thuc Nguyen, Tran Minh The Uyen.

**Formal analysis:** Pham Son Minh, Van-Thuc Nguyen.

**Funding acquisition:** Pham Son Minh, Van-Thuc Nguyen, Tran Minh The Uyen.

**Investigation:** Van-Thuc Nguyen, Tran Minh The Uyen, Thanh Trung Do.

**Project administration:** Nguyen Truong Giang, Pham Son Minh, Tran Anh Son, Tran Minh The Uyen, Van Thanh Tien Nguyen.

**Visualization:** Pham Son Minh, Van-Thuc Nguyen.

**Writing – original draft:** Pham Son Minh, Van-Thuc Nguyen, Tran Minh The Uyen.

**Writing – review & editing:** Pham Son Minh, Tran Anh Son, Van-Thuc Nguyen, Thanh Trung Do.

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
