## [Decision Letter · Decision Letter 0]

18 Mar 2024

PONE-D-24-07210External gas-assisted mold temperature control and optimization molding parameters for improving weld line strength in polyamide plasticsPLOS ONE

Dear Dr. Nguyen,

Thank you for submitting your manuscript to PLOS ONE. After careful consideration, we feel that it has merit but does not fully meet PLOS ONE’s publication criteria as it currently stands. Therefore, we invite you to submit a revised version of the manuscript that addresses the points raised during the review process.

We look forward to receiving your revised manuscript.

Kind regards,

Khalil Abdelrazek Khalil, Ph.D.

Academic Editor

PLOS ONE

Journal Requirements:

Reviewers' comments:

Reviewer's Responses to Questions

**Comments to the Author**

1. Is the manuscript technically sound, and do the data support the conclusions?

Reviewer #1: Yes

Reviewer #2: Partly

2. Has the statistical analysis been performed appropriately and rigorously? 

Reviewer #1: Yes

Reviewer #2: No

3. Have the authors made all data underlying the findings in their manuscript fully available?

Reviewer #1: Yes

Reviewer #2: Yes

4. Is the manuscript presented in an intelligible fashion and written in standard English?

Reviewer #1: Yes

Reviewer #2: No

5. Review Comments to the Author

Reviewer #1: The manuscript lacks a certain depth of research and needs to be revised. There are a number of questions that require detailed answers from the authors：

1. Figure 14 illustrates the microscopic morphology of the sample at the fracture location. It is recommended that SEM observations be made of the fracture locations of all samples to confirm the morphology of the toughness fracture samples by comparison.

In fact, the authors lack a description of the toughness fracture mechanism. What causes ductile fracture? What intrinsic relationship exists with changes in process parameters?

2. In addition, it is recommended that the authors perform liquid nitrogen immersion and quenching of the weld line of samples with and without mold heating, and SEM observations to investigate the effect of the different processes on the weld micro-morphology.

3. Please discuss what at 0.2s packing time, UTS and elongation are not linearly related to the other sample groups. Is it possible that there is a zone of action between not packing time and packing time for 0.4s, such as the effect on the internal stresses in the product? The inappropriate packing time causes it to negatively affect the internal stresses of the product, which in turn leads to a decrease in the mechanical properties of the product? This reason needs to be discussed in depth. Characterization is recommended to verify the causes and inferences.

4. It is very important that authors need to state the principle of setting the filling time. In general, the VP switchover point of the injection molding process is constant, then a longer filling time means a smaller injection speed.

But the author describes it as follows: “A filling period that is too short may result in an incomplete filling, poor part development, and sufficient integrity of the structure. Excessive filling time, on the other hand, might result in overpacking and excessive material, warpage, or part distortion”.

This is clearly wrong. Excessive filling time means slow injections and generally no over-packing.

This paper focuses on the strength of the weld line, and ensuring the same amount of filling is a necessary prerequisite, otherwise there are too many experimental variables to make comparisons.

If the authors used changing the filling time to change the amount of filling, then the whole study is wrong and unsupported. This is because the lower the filling amount, the worse the strength of the weld; the higher the filling amount, the stronger the weld.

Reviewer #2: The authors have explored an injection molding strategy focused on enhancing weld line strength and ductility in PA6 composite samples through elevated mold temperatures. They investigated the impact of different injection molding parameters and conducted further optimization using the ANN-GA approach. However, while the study is intriguing, it lacks clarity, depth, and adequate information in various sections of the manuscript.MAJOR REVISIONS are necessary as detailed below before considering it for publication in the journal.

1. In the abstract, the statement "To optimize the molding parameters, we apply an Artificial Neural Network (ANN) in conjunction with a Genetic Algorithm (GA), marking a sophisticated leap in precision and efficiency over standard optimization methods" requires clarification. Were the results compared with those obtained from other optimization algorithms?

2. “The viscosity of the injection plastic rises, leading to a higher resistance during the molding process.” What resistance? Clarify.

3. The sentence “Besides, increasing the mold temperature by using a mold temperature control system' lacks clarity and completeness. It's unclear what exactly this line signifies. There are multiple sentences of similar nature throughout the manuscript that require clearer expression.

4. "Have the authors prepared Figure 1? If not, it should be properly cited. The figure itself lacks clarity."

5. Was the proposed 'novel Cavity' designed by the authors? Clarify.

"Numerous studies have explored mold temperatures exceeding 100°C. For example, a manuscript published in 2013:

"Meister, S. and Drummer, D.. "Influence of Mold Temperature on Mold Filling Behavior and Part Properties in Micro Injection Molding" International Polymer Processing, vol. 28, no. 5, 2013, pp. 550-557. https://doi.org/10.3139/217.2804"

6. "The novelty of the work remains unclear and should be distinctly highlighted in relation to the presented literature review. Additionally, many sentences in the introduction lack clarity and are incomplete."

7. “The injection mold was designed and manufactured with the core and caity plate as in Figure 2.” Verify the figure number.

8. Why did the author choose not to employ a standard Design of Experiments (DOE) approach in parameter preparation?

9. Indicate the results of UTS and %E in table 1 for better clarity.

10. “The effects of filling time on the tensile test results are shown in Figs. 4, 5.” Correct it to only figure 5 as figure 4 only represent stress strain relationship.

11. “Packing time at 0.2s seems to not enough for an improvement in the UTS value. However, from 0.4s to 0.8s packing time, the UTS value of the PA6 sample experiences an increase compared to sample without packing step.” Why? Provide detailed explanation.

12. The parameter ranges, particularly for melt temperature, are limited to only 2°C. It appears that the defined ranges for each parameter may not be sufficient to draw conclusive results.

13. Include details about the datasets used for training, validation, and testing. Additionally, clarify the strategy employed for selecting datasets for the same.

14. “First, we train the neural network with the input data, which is the UTS value of the samples.” Avoid the use of pronouns such as 'We' in the manuscript. Additionally, clarify whether UTS is considered input data.

15. The modeling and optimization section lacks comprehensive understanding and information. For instance:

a. Crucial details regarding ANN modeling, such as architecture, ANN parameters, epochs, number of hidden layers, and hidden layer neurons, are omitted.

b. The specifics of GA and its parameters are not provided.

16. Table 3 does not adhere to the Pareto principle. The author should familiarize themselves with Pareto optimal solutions.

17. In Table 4, clarify how the variation is calculated. Additionally, explain why the authors did not utilize a standard tool such as ANOVA to determine the influencing parameters.

18. Figure 14 represents which sample? parameter setting?

19. The author's classification of the fracture as ductile raises questions, as flat surfaces typically indicate brittle fractures. Further clarification is needed.

20. “The optimal molding parameters calculated from ANN and GA methods are a filling time of 3.8 s, a packing time is 0.8 s, a melt temperature of 244°C, and a mold temperature of 35.2 °C.” What are the corresponding values for UTS and %E? Has the author conducted any validation experiments?

6. PLOS authors have the option to publish the peer review history of their article (what does this mean?). If published, this will include your full peer review and any attached files.

Reviewer #1: No

Reviewer #2: No

---

## [Author Response · Author response to Decision Letter 0]

23 May 2024

Please read the attachments. Thank you.

---

## [Decision Letter · Decision Letter 1]

8 Jul 2024

External gas-assisted mold temperature control and optimization molding parameters for improving weld line strength in polyamide plastics

PONE-D-24-07210R1

Dear Dr. Nguyen,

We’re pleased to inform you that your manuscript has been judged scientifically suitable for publication and will be formally accepted for publication once it meets all outstanding technical requirements.

Kind regards,

Khalil Abdelrazek Khalil, Ph.D.

Academic Editor

PLOS ONE

Additional Editor Comments (optional):

Reviewers' comments:

Reviewer's Responses to Questions

**Comments to the Author**

1. If the authors have adequately addressed your comments raised in a previous round of review and you feel that this manuscript is now acceptable for publication, you may indicate that here to bypass the “Comments to the Author” section, enter your conflict of interest statement in the “Confidential to Editor” section, and submit your "Accept" recommendation.

Reviewer #1: All comments have been addressed

Reviewer #2: All comments have been addressed

2. Is the manuscript technically sound, and do the data support the conclusions?

Reviewer #1: Yes

Reviewer #2: Yes

3. Has the statistical analysis been performed appropriately and rigorously? 

Reviewer #1: Yes

Reviewer #2: Yes

4. Have the authors made all data underlying the findings in their manuscript fully available?

Reviewer #1: Yes

Reviewer #2: Yes

5. Is the manuscript presented in an intelligible fashion and written in standard English?

Reviewer #1: Yes

Reviewer #2: Yes

6. Review Comments to the Author

Reviewer #1: (No Response)

Reviewer #2: The authors have put forth commendable effort in revising the manuscript. The current version is in an acceptable form, and thus my final decision is to accept it.

7. PLOS authors have the option to publish the peer review history of their article (what does this mean?). If published, this will include your full peer review and any attached files.

Reviewer #1: No

Reviewer #2: No

---

## [Editor Report · Acceptance letter]

15 Jul 2024

PONE-D-24-07210R1 

PLOS ONE

Dear Dr. Nguyen, 

I'm pleased to inform you that your manuscript has been deemed suitable for publication in PLOS ONE. Congratulations! Your manuscript is now being handed over to our production team.

Kind regards, 

on behalf of

Dr. Khalil Abdelrazek Khalil 

Academic Editor

PLOS ONE